# Finite Element Analysis for Biodegradable Dissolving Microneedle Materials on Skin Puncture and Mechanical Performance Evaluation

**DOI:** 10.3390/polym13183043

**Published:** 2021-09-09

**Authors:** Qinying Yan, Jiaqi Weng, Shulin Shen, Yan Wang, Min Fang, Gensuo Zheng, Qingliang Yang, Gensheng Yang

**Affiliations:** College of Pharmaceutical Sciences, Zhejiang University of Technology, 18 Chaowang Road, Hangzhou 310032, China; yqy@zjut.edu.cn (Q.Y.); wengjiaqi0088@126.com (J.W.); sslsii@163.com (S.S.); yanwwz97@163.com (Y.W.); fffmtrophy@163.com (M.F.); a765792548@163.com (G.Z.); qyang@zjut.edu.cn (Q.Y.)

**Keywords:** biodegradable dissolving microneedle, mechanical properties, Young’s modulus, Poisson’s ratio, finite element analysis

## Abstract

In this study, a micro-molding technology was used to prepare the microneedles (MNs), while a texture analyzer was used to measure its Young’s modulus, Poisson’s ratio and compression breaking force, to evaluate whether the MNs can penetrate the skin. The effects of different materials were characterized by their ability to withstand stresses using the Structural Mechanics Module of COMSOL Multiphysics. Carboxymethylcellulose (CMC) was chosen as the needle formulation material with varying quantities of polyvinyl pyrrolidone (PVP), polyvinyl alcohol (PVA) and hyaluronic acid (HA) to adjust the viscosity, brittleness, hardness and solubility of the material. The results of both the experimental tests and the predictions indicated that the hardest tip material had a solids content of 15% (w/w
) with a 1:2 (w/w) CMC: HA ratio. Furthermore, it was shown that a solid content of 10% (w/w) with a 1:5 (w/w) CMC: PVA ratio is suitable for making patches. The correlation between the mechanical properties and the different materials was found using the simulation analysis as well as the force required for different dissolving microneedles (DMNs) to penetrate the skin, which significantly promoted the research progress of microneedle transdermal drug delivery.

## 1. Introduction

Nowadays, the most commonly used drug delivery methods are injections and oral treatments, but they both have certain limitations. Oral administration suffers from the first-pass effects of the gastrointestinal tract and liver, causing inadequate drug absorption and low bioavailability. While injections are often accompanied by pain, low patient compliance and/or needle contamination [1]. Microneedles (MNs), a new transdermal drug delivery system, has many advantages, including high patient compliance, no liver first-pass effect [2], controllable drug release and desirable simplicity of operation [3]. It is often applied to deliver biomacromolecules that are not suitable by the traditional oral way, including vaccine antigens, adenovirus vector [4], insulin [5], as well as low molecular weight heparin [6], etc.

MNs are small, miniature wound invasive devices with a needle length of 25–2000 μm and a tip diameter of 1–25 μm, which can easily penetrate the dermis to deliver drugs without pain and with a very low probability of bacterial infection [7]. They can be generally divided into solid microneedles [8], coated microneedles [9], hollow microneedles [10], hydrogel-forming microneedles and dissolving microneedles (DMNs) [11]. Generally, DMNs are fabricated with various high molecular polymers such as chitosan (CS) [12], poly-lactide-co-glycolide (PLGA) [13], polymethyl methacrylate (PMMA) [14], CMC [15], PVA [16], PVP [17] and HA [18], due to their biocompatibility, biodegradability, tougher and higher solubility than other materials [19]. It is generally known that different polymer materials have different hardness, solubility and plasticity. Therefore, it is difficult for single-component polymer DMNs to have the highest hardness, solubility and plasticity simultaneously. On the contrary, since the multicomponent polymer DMNs contained a variety of materials in different proportions, the mechanical properties such as hardness, solubility and plasticity of the DMNs can be easily adjusted by fine-tuning the type, proportion and water content of the composite materials of DMNs.

The skin can be divided into two parts: the epidermis and the dermis. The dermis layer contains abundant water, nerve endings, capillaries, capillary lymphatic vessels and receptors; most drugs can directly act on the target receptor in the dermis layer to exert a therapeutic effect. The stratum corneum in the epidermal layer has a specific hindrance effect that prevents drugs from entering the systemic circulation, therefore affecting the efficacy of the drug [20]. The mechanical behavior of the DMNs is one of the most significant considerations in several clinical applications. It determines whether the DMNs possess sufficient mechanical properties to pierce the dermal papilla layer and absorb moisture from the skin to dissolve and release the loaded drug.

As mentioned above, the mechanical properties of multi-component DMNs were significantly affected by the type, proportion and water content of the DMNs materials. However, there are a large number of existing DMNs materials, which means that it will take a relatively long time and a relatively high economic cost to analyze the influence of the material on the mechanical properties of the DMNs only through actual experimental measurement. Therefore, to understand the mechanical behavior of DMNs, it is proposed to analyze the minimum force of penetration into the skin and the max-penetration depth through finite element analysis simulation. Finite element analysis (FEA) is software that simulates real physical systems (geometry and load cases) using mathematical approximations. With interacting and straightforward elements (ie., units), a finite number of unknowns can be used to approximate an infinitely unknown real system. What if the mechanical properties of the DMNs are simulated by the basic properties such as Young’s modulus and Poisson’s ratio of the polymer material, a large number of single or mixed materials can be screened before the actual fabrication and test. It will possibly be a simple and efficient way to find the most suitable polymers for preparing DMNs.

The present study aims to establish a method that can simulate the mechanical properties of DMNs and find more prescriptions for DMNs that can successfully penetrate the human dermis for drug delivery by this method.

## 2. Materials and Methods

### 2.1. Materials and Animals

Carboxymethylcellulose sodium (CMC-Na) and polyvinyl alcohol (PVA) were purchased from Aladdin (Shanghai, China), hyaluronic acid (HA) was purchased from Freda (Shandong, China) and polyvinyl pyrrolidone (PVP) was purchased from Boao Biotechnology Co. Ltd. (Shanghai, China). Five weeks-old inbred JCR male mice (20–23 g) were purchased from the Zhejiang Academy of Medical Sciences (Hangzhou, China). All studies were reviewed and approved by the Animal Care and Use Committee. All studies were reviewed and apprived by the Institutional Animal Care and Use Committee at Zhejiang University of Technology (20190301007). The study was conducted according to the guidelines of the Declaration of Helsinki, and approved by the Ethics Committee of Shenzhen Second People’s Hospital (20200601018-FS01).

### 2.2. Preparation of the Composite Polymer Material

Deionized water was added to different combinations of mixed polymers, then put on a four-dimensional rotary mixer for 12 h to obtain an evenly mixed solution, resulting in a 15% (w/w) aqueous mixture for the needles, and a 10% (w/w) aqueous mixture for the patches. Both were then sealed and placed in a refrigerator at 4 °C for storage.

### 2.3. Preparation of the DMNs

For this study, microneedle male mold, a uniform and sharp silicon-made main structure of needles, 6 × 6 (with 500 μm height, 130 μm width at base, 12 μm width at the tip, 1500 μm interspacing between the needles), was created using an engraving machine. Silicone elastomer base and silicone elastomer curing agent were mixed with a 10:1 (w/w) ratio while the main structure was placed in polydimethylsiloxane (PDMS) and cured at 70 °C to product microneedle female mold. Then, the main structure was carefully peeled off, thus obtaining the inverse replicated micro molds. To evaluate the effect of the different materials on the DMNs’ mechanical properties as shown in Figure 1. Three polymer materials were chosen: HA, PVP and PVA mixed with CMC in different proportions as shown in Table 1.

0.5 g of the prepared needle material was added into the PDMS micro mold and centrifuged at 3000 rpm for 5 min, one way, then centrifuged at 3000 rpm for 5 min in the other direction. The process was repeated three times to pressure the gel into the cavities, after which the excess material was removed with a pipette and recycled. Subsequently, the filled micro molds were placed in the vacuum box for 12 h. 0.3 g of the prepared patch material was then added to the PDMS mold and the centrifuge process was repeated as above. Once complete, the micro molds were dried at room temperature for 2 days to obtain the complete DMNs. The dry molded DMNs were then carefully removed with sharp tweezers and sealed for storage.

### 2.4. Mechanical Properties of DMNs

#### 2.4.1. Tensile Tests

The three materials (CMC/HA, CMC/PVP and CMC/PVA) were prepared into 15% (w/w) aqueous uniform thickness films and pressed into a “dog bone” shape using a national standard sample model. The “dog bone” sample was then dried at room temperature for 2 days. The average thickness of the “dog bone” sample film was 0.05 mm. The film was subjected to a mechanical tensile test using an A/TG probe at a constant speed until breakage occurred. First, the Stretch model was chosen, the A/TG probe set at a motion speed of 0.1 mm/s, the displacement set to 3 mm, the trigger force, to 0.049 N, and the data acquisition rate to 50 while keeping the probe and the film parallel to each other. The displacement slowly rose until the film was fully deformed. The analyzer was used to record the probe’s tension during the test yielding a force-time curve. The stress and strain were calculated according to Equations (1) and (2), and the stress-strain curves were plotted. The ultimate tensile strength is obtained when the sample breaks as it represents the maximum load a sample can withstand, and the maximum elongation is its corresponding displacement. Young’s modulus (Et), the stress-strain curve within the elastic deformation range, was calculated according to Equation (3). Poisson’s ratio (υ) of the composite material was calculated according to Equation (4).
(1)$$\mathrm{Stress}=\frac{F}{W\times T}$$
(2)$$\mathrm{Strain}=\frac{\Delta L}{L}$$
(3)$$Et=\frac{F\times L}{A\times \Delta L}$$
(4)$$\upsilon =-\frac{\Delta W/W}{\Delta L/L}$$
where F is the applied tensile force during the stretching process, W, L and T are the sample’s initial width, length and thickness, respectively, and A is the sample’s cross-sectional area, ΔW is the change in width, and ΔL is the change in length.

#### 2.4.2. Hardness Tests

The prepared DMNs’ patches were cut into 6 × 6 pieces, fixed onto a 3 mm thick PVC plate using tape, and placed on the aluminum base of the texture analyzer to prevent causing damage to the texture analyzer’s probe due to direct collision with the base. The compression mode was then chosen, the p/6 probe set to a movement speed of 0.1 mm/s, with a displacement of 500 μm, a trigger force of 0.049 N and a data acquisition rate of 50. The probe and DMNs were kept axially parallel during the compression, and the analyzer recorded the probe’s pressure until the DMNs were fully deformed. To visually show the effect of different forces on the mechanical properties of the DMNs, the force of 0–0.436 N was applied to the single microneedle of optimal prescription, the camera recorded the DMNs morphology changes under different stress conditions, and the length reductions of the DMNs were obtained by calculation.

### 2.5. Simulation of DMNs

#### 2.5.1. Structure Model Construction

Simulations were performed on a single cone-shaped microneedle with 500 μm height, 130 μm base diameter and 12 μm tip diameter using the Structural Mechanics Module of COMSOL Multiphysics (COMSOL Inc., Shanghai, China). The 3D base shape of a single microneedle was created using the polygonal tool, and the linear elastic material was chosen as the DMNs material, then extruded 500 μm in the Z-axis with scale factors in the X and Y directions of 0.06, yielding a fixed tip diameter of 12 μm linearly in COMSOL multiphysics. The single needle structure was modeled as a linear elastic material by measuring Young’s modulus and Poisson’s ratio of each mixed material.

#### 2.5.2. Von-Mises Stress

In this paper, a cylinder with a diameter of 300 μm is selected to simulate the epidermis, and three different heights of the cylinder sections are chosen to simulate the stratum corneum, the living epidermis and the dermis, which Young’s modulus and density were 7.33 Mpa, 0.752 Mpa, 0.489 Mpa and 1300 kg/m^3^, 1300 kg/m^3^, 1200 kg/m^3^, respectively. The height of the stratum corneum is 20 μm; the height of the active epidermis is 85 μm; the height of the dermis is 1000 μm. The skin structure is assumed to be an incompressible hyperelastic model (Neo-Hooken model) with parameters C_10_ of 10 Mpa, 0.16 Mpa and 0.16 Mpa for each layer, respectively [15]. Both the epidermis and the dermis are considered incompressible materials with a Poisson’s ratio of 0.48. The structural mechanics module of COMSOL Multiphysics is used to define stress analysis, using static analysis to simulate Von-Mises stress and microneedle/skin deformation under static equilibrium. Applying a fixed constraint on the bottom surface of the dermis, microneedle movement only in the axial direction. The force range was increased from 0.01 to 0.5 N, and the friction coefficient between the microneedle and the skin was set to 0.42. Measure the displacement of the DMNs under different forces, and the corresponding force was recorded as complete penetration force when the displacement is 500 μm. Applying a fixed constraint on the tip of the microneedle, microneedle movement only in the axial direction on. A 0.1 N axial load force was applied on the bottom of the microneedle, and the max-penetration depth of the microneedle was obtained by simulation calculation.

### 2.6. Swelling Tests

The completely dried DMNs were placed in a 70 °C oven (Senxin, Experiment Instrument Co., Ltd, Shanghai, China). to dry until the mass remains at the same value after consecutive three times and, the dry DMNs were placed on the perforated PE film inversely ensuring that only the needles are exposed to a pH 5.5 PBS buffer (simulated skin pH), placed in a 37 °C, 50 rpm full temperature shaking incubator Swelling test. The DMNs were taken every 10 s, where the water was blotted with filter paper, and the swelling of the DMNs was observed using an inverted microscope and photographed.

### 2.7. Skin Preparation

In this case, 5-weeks-old mice were purchased and fed accordingly for 2 weeks. Then their back hair was removed using a razor first and the remaining hair was carefully removed use an electric shaver (chemical hair removal is not used to protect the integrity of the skin tissue) after euthanizing them by removing their vertebrae. The layer of fat and blood vessels of the isolated skin was wiped off with absorbent cotton, and the skin was rinsed with physiological saline, and preserved in saline at 4 °C.

### 2.8. Skin Tests

#### 2.8.1. Isolated Skin Penetration

The prepared DMNs patches (6 × 6) were fixed onto the P/6 probe and the mice skin without subcutaneous fat and blood vessels were placed onto a 3 mm thick PVC which plated on the aluminum base of the texture analyzer (to prevent causing damage to the texture analyzer’s probe due to direct collision with the base). The compression mode was then chosen, the P/6 probe set to a movement speed of 0.1 mm/s, with a displacement of 100, 200, 300, 400, 500 μm, a trigger force of 0.049 N and a data acquisition rate of 50. The probe and DMNs were kept axially parallel during the compression, and the analyzer recorded the probe pressure. After removing the DMNs, place the puncture surface skin down and place it on the bottom surface of the trapezoidal prism of optical coherence tomography(OCT) [21], to measure and calculate the depth of penetration of the microneedle into the skin. After the measurement, the punctured skin surface was exposed to 0.4% (m/v) trypan blue solution for 10 min, and the excess coloring was washed with alcohol. The isolated skin was tiled in a 4% paraformaldehyde tissue fixative for 24 h to harden the tissue for subsequent sectioning. Alcohol is used as a dehydrating agent, increasing from a low concentration to a high concentration, gradually removing moisture from the isolated skin, and the skin was placed in xylene to make it transparent. After paraffin wax was completely immersed in the isolated skin, the paraffin block embedded with the isolated skin is fixed on the microtome, cut into 5 μm thick slices, and stained with the hematoxylin-eosin method (H&E staining), so that the excised skin tissue can be observed more clearly under the microscope.

#### 2.8.2. Living Skin Irritation and Rehabilitation

The hair removal 5-weeks-old mice were selected, and the 500 μm-length DMNs were administered perpendicular to the surface of the back skin with a hammer for 30 s, then removed. The irritation of the mice skins was observed at regular intervals according to the Draize [22] Skin Reaction Judgment Criteria Score (Table 2).

The female volunteer’s wrist’s skin (soft and flat enough) was selected, and the 500 μm-length DMNs were administered perpendicular to the surface of the skin with a hammer for 30 s, then removed. The healing of the wound was observed at intervals and photographed.

## 3. Results and Discussion

### 3.1. Simulation

Buckling is defined as the loss of structural stability and mechanical behavior of the DMNs. Through the finite element simulation diagram (Figure 2a), we can find that after a certain pressure is applied to a single microneedle, the force value of the tip of the microneedle can reach 5 times that of other parts. Therefore, during the buckling process of the microneedle, the tip fracture is the main reason leading to the loss of mechanical behavior of the DMNs. The compression breaking force which is a value for evaluating whether the needle tip is broken or not, can be detected through the texture analyzer and indicate the force at the moment of buckling the loss of mechanical properties of the DMNs. As shown in Table 3, multi-component polymer microneedles composed of different materials have different compression breaking forces, and it is closely related to Young’s modulus.

Von-Mises stress can refer to the principal stresses in the x, y, z dimensions, which always be used to predict the mechanical properties of DMNs. The penetration force calculated by Von-Mises stress is accustomed to predicting the force required for skin penetration. In this study, we use the finite element method to simulate the whole process of a single microneedle penetrating the skin. It can be seen from Figure 2b that both the microneedle and the skin move down to a certain extent during the penetrating process, which indicates that the penetrating behavior is indeed taking place. In addition, it can also be easily seen combined with the maximum penetration depth of the skin, it can be found that the stronger the mechanical properties of the DMNs, the deeper the penetration depth.

In previous studies, the penetration force is often used as a criterion for judging whether the puncture behavior is successful. However, in this study, there is an obvious thing that can be found, no matter the proportions or materials of the DMNs change, the penetration force is always constant, around 0.048 N, which indicates that the penetration force does not change with the variation of the proportions or materials of the DMNs. On the contrary, the Von-Mises stress rises with the increase of Young’s modulus. The phenomenon does not mean that as long as the DMNs have identical shape and size, their role in the penetration behavior is the same. By comparing compression-breaking force and penetration force in Table 3, it can be found that the value of compression breaking force is much larger than penetration force, which proves that all DMNs can easily penetrate the skin and penetrate the dermis, theoretically. However, in actual experiments, it was found that there are still some DMNs buckling during the process of skin penetration. Three groups of DMNs have greater resistance in the initial stage of the process of penetrating the skin, and after pulling out the DMNs, it was found that the tips have buckled. It can be found from the values in Table 3 that Young’s modulus of these three groups of DMNs are smaller than that of the other groups, and the maximum penetration depth can only reach about 200 μm, which makes them hard to reach the dermal layer.

To sum up, it is not enough to evaluate the puncture behavior of the DMNs by simply comparing the values of compression breaking force and penetration force. The two groups of DMNs with the same compression-breaking force may have different Von-Mises stress due to their different proportions or materials, thus have completely different piercing behaviors. However, the actual measurement process of various data including compression breaking force, Von-Mises stress, penetration force and max-penetration depth is difficult, and the repetition rate is low, due to the soft and uniformly thick puncture object. To measure the value with small errors, dozens of measurements are required, which consumes a lot of time and money. It is very ideal and efficient if an accurate and efficient simulation method can be used for screening before the actual experiment. The max-penetration depth in Table 3 can predict the penetration depth of DMNs, and comparing with the actual penetration depth measured through the OCT, it can be found that the experimentally actual penetration depth substantially coincides with the simulated max- penetration depth. Therefore, we can boldly speculate that the model used in this study is ideal for the simulation of skin penetration behavior.

### 3.2. Puncture Tests

Through actual experimental measurement and simulation, we found that the hardest tip material had a solids content of 15% (w/w) with a 1:2 (w/w) CMC: HA ratio, its compression breaking force can reach 0.48N, and it can easily reach a max-penetration depth of 500 μm to achieve dermal drug delivery. In order to investigate the mechanical properties and drug delivery potential of the DMNs of this prescription in more detail, we selected this prescription as the optimal prescription for follow-up experiments. Observing the OCT image of the puncture site (Figure 3a,b), 6 clear pinholes with a depth of 500 microns can be clearly found after skin puncture, caused by a single row of microneedles piercing the skin. Histology images of the puncture site (Figure 3c) show that the DMNs has a continuous cavity (around 500 μm) from top to bottom. This phenomenon indicates that the DMNs penetrates the epidermis and reaches the dermis layer, proving that DMNs have sufficient length and hardness to acquire the dermis layer, achieving an efficient transdermal drug delivery system.

### 3.3. DMNs Images and Characterization

The DMNs images were taken using light microscopy (Figure 4a), and SEM (Figure 4b,c). Figure 4a shows that the needles are sharp, with a smooth surface, without offset and skew, and the patch is flat enough. Figure 4b,c show that DMNs with a height, width at the base, width at the tip, and interspacing between needles of 500 μm, 130 μm, 12 μm, 1500 μm, respectively.

### 3.4. Structural Analysis

The mechanical properties of DMNs were characterized using a texture analyzer, through which the composite’s Young’s modulus (Et) and Poisson’s ratio (υ) were calculated (Table 3). Young’s modulus represents the ability of a material to resist deformation, i.e., as the modulus increases, the material stiffness also increases. It was found that for a CMC:HA ratio of 1:2 (w/w), the Single microneedle hardness is maximal, reaching 0.48 N (Figure 5a). Tensile breaking force represents the deformation resistance of DMNs patch backing layer, as shown in Table 3, CMC:PVA ratio of 1:5 (w/w) and CMC:HA ratio of 1:2, having high tensile breaking force, compared to other groups. It was found that for a CMC:PVA ratio of 1:5 (w/w), the elongation is longer (1.36 mm), which means it has relatively higher toughness and is more suitable to be used as a backing layer for DMNs after solidification of the patch. As shown in Figure 5b, the reduction of the single DMN below 10%, and the mechanical properties remain intact when 0.005 N force was subjected to the DMN, which had a solids content of 15% (w/w) with a 1:2 (w/w) CMC: HA ratio. Access to literature, the force of 0.005 N is much larger than 3.596 × 10^−4^ N [23] that required for the microneedle to penetrate the skin. The microneedle had no visible bending and had good mechanical properties when it received 3.596 × 10^−4^ N. The reduction of the single DMN reached 50% when a single DMN was subjected to a pressure of 0.436 N. All experimental data described that the DMN can easily penetrate the skin for administration under the condition of its sufficient mechanical properties.

### 3.5. Swelling Tests

The wholly dried DMNs were placed upside down on the perforated PE film and exposed to a pH 5.5 PBS buffer. As time goes by, the DMNs gradually swelled and then dissolved in the water. As shown in Figure 6 when DMNs were exposed to the PBS buffer at first, the tip of the DMN started swelling. 20 s later, the whole DMN began swelling and dissolving down to 1/3. 30 s later, DMN dissolving down to 1/4. Finally, DMN completely dissolved after 40 s. Swelling test results showed that the DMNs have good solubility, thus can absorb moisture from the skin to dissolve and release the loaded drug fast when inserted into the skin.

### 3.6. Skin Penetration Studies

#### 3.6.1. Isolated Skin Penetration

The DMNs were administered perpendicular to the surface of the skin with a hammer for 30 s for a complete puncture (Figure 7a). After removing the DMNs, the punctured part of the skin was exposed to 0.4% (m/v) trypan blue solution (Figure 7b). Trypan blue is a cell-reactive dye that can be used to assess cell membrane integrity. After dyeing, the pinholes produced by the DMNs were visible and light blue, indicating that the DMNs had not only created indentations on the skin but pierced the skin causing damage to the cell membrane. Thus, the prepared DMNs were sufficiently enabled to penetrate the skin and deliver the drug.

#### 3.6.2. Living Skin Irritation and Rehabilitation

The 500 μm-length DMNs were administered perpendicular to the surface of the mice’s back skin. Next, the irritation of the mice skins was observed and scored at regular intervals according to the Draize Skin Reaction Judgment Criteria Score Table after DMNs were removed. As shown in Table 4, the skin was slightly erythema when the DMNs were removed, and as time passed, the skin becomes no erythema very quickly (about 7 min). The comprehensive irritation score was 0.347, which means that the DMNs were non-irritating to the skin. In addition, this conclusion can also be obtained in the living skin rehabilitation experiment (Figure 8).

The DMNs penetrated the wrist skin of volunteers, where the skin is flat and easy to observe. They were applied for 30 s after a complete puncture into the skin. The skin healing condition was observed and recorded with a camera (Figure 8). It was observed that the skin pinhole is visible at the very beginning. After 240 s, the skin pinhole became blurred. 360 s later, the skin pinhole almost disappeared. In addition, after 30 min, the skin was fully healed.

## 4. Conclusions

Due to the receptors in the skin allow transdermal administration to act on the target sites directly, DMNs can also greatly shorten the onset time of drugs and reduce the dosage of drugs, which has a great effect on avoided side effects. In this study, we combine finite element simulation with actual experiment, found that DMNs with identical shapes and sizes have the same penetration force but have different penetration behaviors. The penetration behaviors of DMNs are controlled by their mechanical properties, and the mechanical properties of DMNs are controlled by the proportions or materials of the DMNs. This is the beginning, next, we can explore the influence of DMNs’ shape, size, and the distance between the needles; can explore the application and optimization of DMNs in different indications, races, and patients of different ages, provide essential guidance for further microneedle transdermal administration.

## Figures and Tables

**Figure 1 polymers-13-03043-f001:**
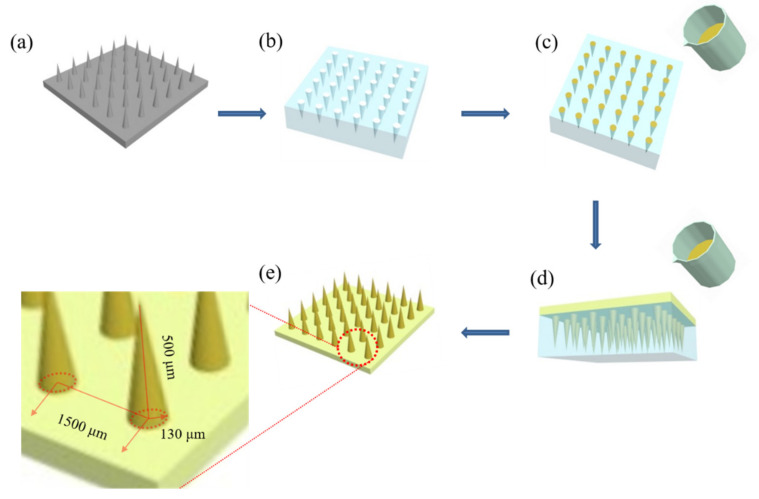
(**a**) Microneedle male mold; (**b**) Microneedle female mold; (**c**) Process of tip preparation; (**d**) Process of making patch preparation; (**e**) DMNs patch.

**Figure 2 polymers-13-03043-f002:**
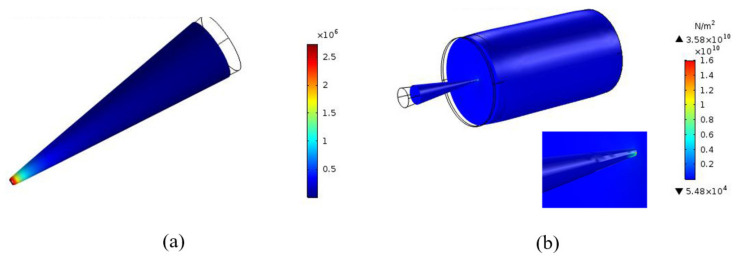
(**a**) Critical buckling force prediction for single DMN; (**b**) Von-Mises stresses prediction for single DMN (The color in the picture is related to the force, the closer the color is to red, the greater the force).

**Figure 3 polymers-13-03043-f003:**
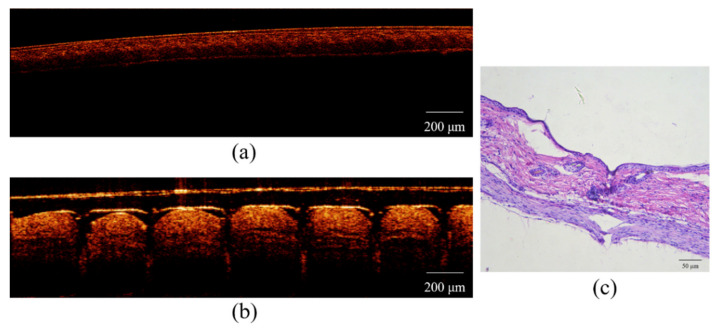
(**a**) OCT image before skin puncture; (**b**) OCT image after skin puncture; (**c**) histology images of the puncture site.

**Figure 4 polymers-13-03043-f004:**
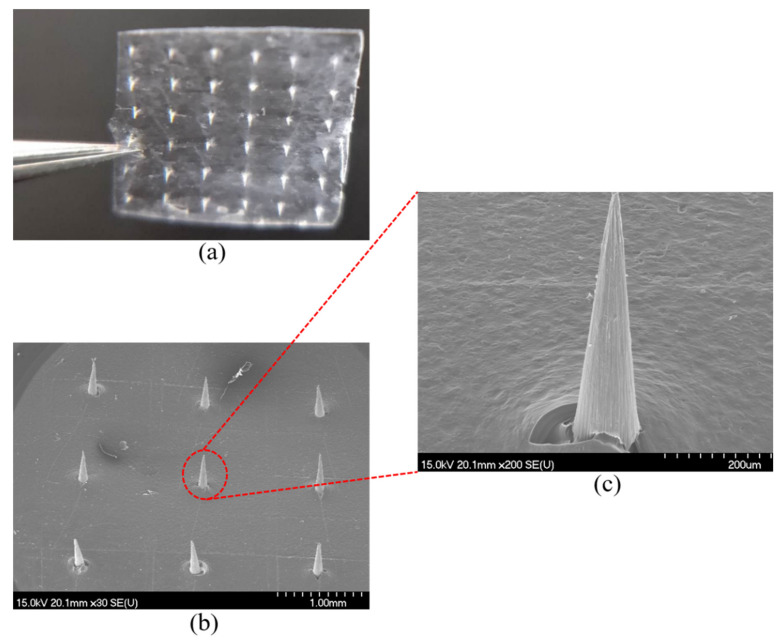
(**a**) DMNs’ image of light-microscopy; (**b**) DMNs’ image of SEM; (**c**) DMNs’ partial enlarged image of SEM.

**Figure 5 polymers-13-03043-f005:**
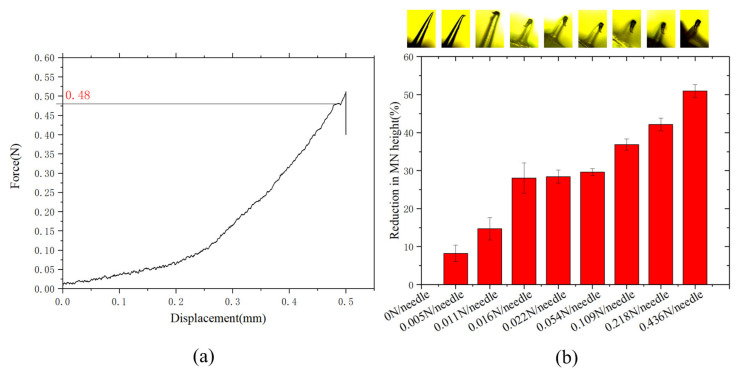
(**a**) The stress curve of a microneedle; (**b**) The morphology through a microscope and the reduction of the microneedle change under different stress conditions.

**Figure 6 polymers-13-03043-f006:**
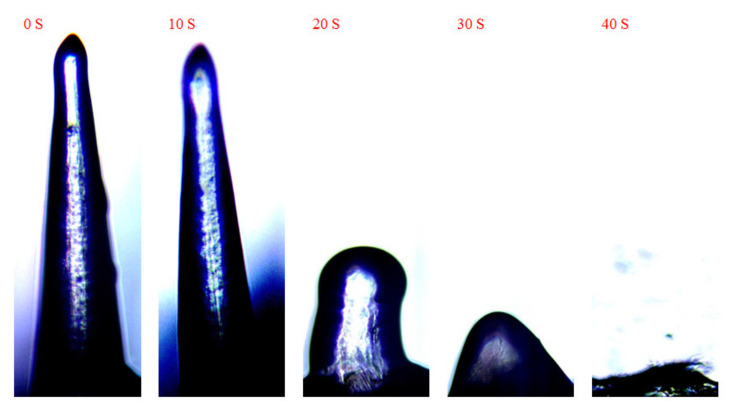
Swelling change of DMNs over time after exposure to water.

**Figure 7 polymers-13-03043-f007:**
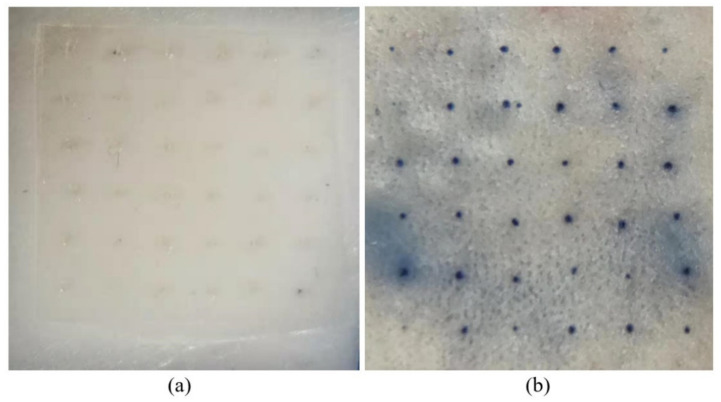
(**a**) DMNs piercing into the isolated skin; (**b**) Isolated skin acupunctured part was stained using trypan blue solution (Since trypan blue only stains dead cells, the appearance of blue indicates that the skin cells are punctured and inactivated).

**Figure 8 polymers-13-03043-f008:**
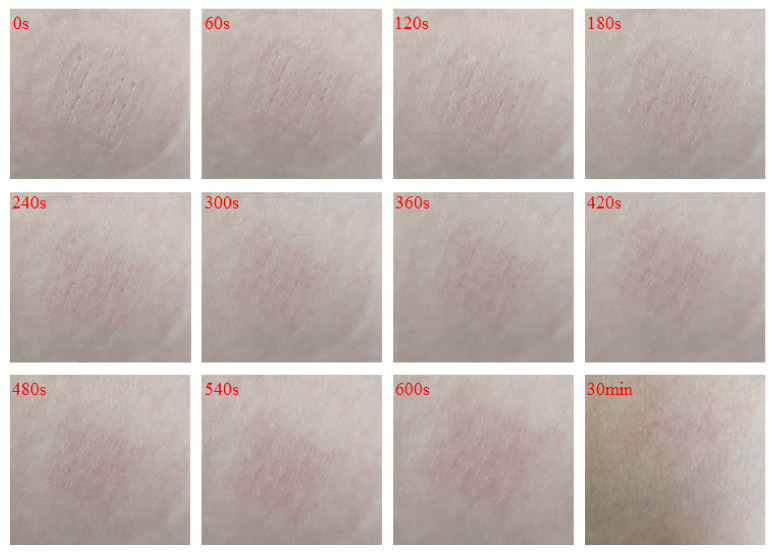
Record the healing of the DMNs-punctured skin wounds over time through continuous video recording, and take screenshots every 60 s to record.

**Table 1 polymers-13-03043-t001:** CMC mix with different proportions of three polymer materials: HA, PVP, PVA.

Composition	Proportion
CMC-PVP	CMC:PVP = 1:2
	CMC:PVP = 1:5
	CMC:PVP = 2:1
	CMC:PVP = 5:1
CMC-HA	CMC:HA = 1:2
	CMC:HA = 1:5
	CMC:HA = 2:1
	CMC:HA = 5:1
CMC-PVA	CMC:PVA = 1:2
	CMC:PVA = 1:5
	CMC:PVA = 2:1
	CMC:PVA = 5:1

**Table 2 polymers-13-03043-t002:** Draize Skin Reaction Judgment Criteria Score.

Erythema and Eschar/Formation	Value
No erythema	0
Very slight erythema (barely perceptible. Edges of area not well defined)	1
slight erythema (pare red in color and area well defined)	2
Moderate to severe erythema (defined in color and area well defined)	3
Severe erythema (beet to crimson red) to slight eschar formation (injures in depth)	4

Skin Irritation score PII = Σ (1, 3, 5, 7, 10 min)/(Number of mice (8) × Irritation site (1) × Number of observations (3)).

**Table 3 polymers-13-03043-t003:** The mechanical property characterization and simulated value of different composite material.

	CMC:PVA				CMC:PVP				CMC:HA			
	1:5	5:1	1:2	2:1	1:5	5:1	1:2	2:1	1:5	5:1	1:2	2:1
Tensile breaking force (N)	22.5	5.6	14	8.5	5.1	12.7	11	11.2	9.7	5.3	33.2	8.8
Elongation (mm)	1.36	0.35	1.42	2.78	0.24	0.36	0.62	0.48	0.43	1.30	0.48	0.24
Young’s modulus (Mpa)	5454	2206	5514	1457	2912	5934	2710	3712	3370	1351	10853	4783
Poisson’s ratio	0.40	0.04	0.37	0.08	0.22	0.32	0.21	0.30	0.12	0.43	0.33	0.12
Compression breaking force (N)	0.29	0.15	0.29	0.10	0.19	0.31	0.19	0.23	0.21	0.11	0.48	0.27
Von-Mises stress (Mpa)	555	192	581	142	263	555	246	359	300	116	980	413
Penetration force (N)	0.049	0.047	0.049	0.046	0.049	0.049	0.048	0.048	0.048	0.047	0.048	0.048
Max-penetration depth (μm)	469	214	500	164	306	500	285	398	336	108	500	459
Actual penetration depth (μm)	477	285	458	262	328	469	317	391	395	202	448	455

**Table 4 polymers-13-03043-t004:** Scores of Draize Skin Irritation.

	Mice’ Serial Number	1	2	3	4	5	6	7	8
Time (min)									
1		2	2	2	2	1	2	2	2
2		1	1	1	1	1	1	1	1
3		1	0	0	0	0	1	0	0
5		0	0	0	0	0	0	0	0
10		0	0	0	0	0	0	0	0

PII = 0–0.4 means irritation; PII = 0.5–1.9 means slight irritation; PII = 2.0–4.9 means mild irritation; PII = 5.0–8.0 means severe irritation.

## Data Availability

The data presented in this study are available on request from the corresponding author.

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
