# Peer review of "Finite Element Analysis for Biodegradable Dissolving Microneedle Materials on Skin Puncture and Mechanical Performance Evaluation"

_polymers, 2021, doi:10.3390/polym13183043_

Round 1
Reviewer 1 Report
Due to the receptors in the skin allow transdermal administration to act on the target sites directly, dissolving microneedles (DMNs) can also greatly shorten the onset time of drugs and reduce the dosage of drugs, which has a great effect on avoided side effects. In this study, the authors combine finite element simulation with actual experiment, found that DMNs with identical shapes and sizes have the same penetration force but have different penetration behaviors. The penetration behaviors of DMNs are controlled by their mechanical properties, and the mechanical properties of DMNs are controlled by the proportions or materials of the DMNs. This work could attract a broad reader's interests, would be recommended for the acceptance in Polymers. There are several questions below,
(1) Penetration force would also be related to needle and skin properties, and how do the author explain this?
Reviewer 2 Report
This work by Yan et al., aims to evaluate the mechanical properties of degradable microneedles. The authors test different materials and provide simulation analysis by COMSOL analysis. In general, the article is a straightforward approach to studying microneedles, performing degradation, response to mechanical force, and testing using skin model. The main limitation is that this work doesn’t clearly show anything novel in regards to microneedle design, material, or capabilities, moreover, there are problems with scholarly presentation as most figure captions lack context or information. For instance, dmn is never defined in main text, (although by context is understandable that the authors mean degradable). My biggest concern relies on the limited experimental comparison between different materials, lacks of proper control studies (eg CMC alone) and the statistics presented in the manuscript are not very robust, using a low number of replicas or none at all.
The authors have an interesting idea, but the results do not fully support the article title and conclusions. this could be the basis of an interesting work, but the authors would require validating that their use of a COMSOL module can lead to an actual prediction. This would entail performing tests for all the material combinations and analytically evaluate. For these reasons, I don’t support the publication of this work at this stage.
Reviewer 3 Report
Ethical approval number and date should be included in the manuscript for using animals.
The weight of animals used should be included in the manuscript.
Author should use better word than “killing”.
Author should include how the animals were euthanized.
How the hair were removed from animals.
Ethical approval was obtained or not, for the female volunteer’s wrist’s skin. If yes approval number should be included.
Authors should include skin irritations data.
Round 2
Reviewer 2 Report
The authors did not address any of my comments, in particular, there is no analytical work comparing all the types of microneedles to validate the simulation, making the use of the comsol model not uselfull or novel,
the authors have limited to nonexisting experimental comparison between different materials, lacks of proper control studies (eg CMC alone) and the statistics presented in the manuscript are not very robust, using a low number of replicas or none at all
The authors have an interesting idea, but the results do not fully support the article title and conclusions. this could be the basis of an interesting work, but the authors would require validating that their use of a COMSOL module can lead to an actual prediction. This would entail performing tests for all the material combinations and analytically evaluate. For these reasons, I don’t support the publication of this work at this stage.
Reviewer 3 Report
Authors have revised the manuscript as per the reviewer comments. My recommendation is to accept the manuscript for publication.
